# *Spodoptera frugiperda* (Lepidoptera: Noctuidae) Life Table Comparisons and Gut Microbiome Analysis Reared on Corn Varieties

**DOI:** 10.3390/insects14040358

**Published:** 2023-04-04

**Authors:** Jungwon Jeon, Md-Mafizur Rahman, Changhee Han, Jiyeong Shin, Kyu Jin Sa, Juil Kim

**Affiliations:** 1Interdisciplinary Graduate Program in Smart Agriculture, Kangwon National University, Chuncheon 24341, Republic of Korea; 2Agriculture and Life Sciences Research Institute, Kangwon National University, Chuncheon 24341, Republic of Korea; 3Department of Biotechnology and Genetic Engineering, Faculty of Biological Science, Islamic University, Kushtia 7003, Bangladesh; 4Department of Plant Medicine, Division of Bio-Resource Sciences, College of Agriculture and Life Science, Kangwon National University, Chuncheon 24341, Republic of Korea

**Keywords:** *Spodoptera frugiperda*, FAW, maize, *Enterococcus*, cultivars, microbiome

## Abstract

**Simple Summary:**

The food preference of an insect host plant affects its life cycle. The dietary needs and unique qualities of plants that act as hosts have an impact on insect pest population. Although *Spodoptera frugiperda* prefers maize (*Zea mays*), alternative crops can serve as appropriate hosts if maize crops are not easily available. Considering the six *Z. mays* cultivars in three different categories, “B: Heukjeom 2-ho” was found to be significantly preferred [F (5174) = 19.817; *p* = 0.0001] during the larval rearing stage. A metagenomic analysis of biological triplicate samples (6 × 3 = 18, 5th instar larvae reared on six corn cultivars) revealed that Firmicutes was the most prevalent bacterial community, and Proteobacteria was the second leading bacterial phylum. In terms of the bacterial genera, *Enterococcus* was the most prevalent, followed by *Ureibacillus*. To prevent the spread of this invasive pest, it is vital to gain a comprehensive understanding of FAW’s feeding preferences and gut microbial community composition.

**Abstract:**

The fall armyworm (*Spodoptera frugiperda*, FAW) is an invasive migratory pest that has recently spread to Korea, damaging several corn cultivars with significant economic value. Comparisons of the growth stages of FAW were conducted based on the preferred feed. Therefore, we selected six maize cultivars, including three categories: (i) commercial waxy corn (mibaek 2-ho, heukjeom 2-ho, dreamoak); (ii) popcorn (oryun popcorn, oryun 2-ho); and (iii) processing corn (miheukchal). A significant effect was observed during the larvae period, pupal period, egg hatching ratio, and larvae weight, whereas the total survival period and adult period did not show significant variation among the tested corn cultivars. We identified variations in the FAW gut bacterial community that were dependent on the genotype of the corn maize feed. The identified phyla included Firmicutes, Proteobacteria, Actinobacteria, and Bacteroidetes. Among these genera, the most abundant bacterial genus was *Enterococcus*, followed by *Ureibacillus*. *Enterococcus mundtii* was the most abundant among the top 40 bacterial species. The intergenic PCR-based amplification and gene sequence of the colony isolates were also matched to the GenBank owing to the prevalence of *E. mundtii*. These results showed that the bacterial diversity and abundance of particular bacteria in the guts of FAWs were influenced by the six major maize corn cultivars.

## 1. Introduction

The fall armyworm (FAW) is native to the subtropical and tropical western hemisphere regions of the Americas and is a polyphagous, highly migratory and invasive pest [1]. Although it is native to the Americas, it was identified in West Africa in 2016, and has since expanded its habitat, having now been confirmed to occur in most African countries [2,3] and Europe [4,5]. Recently, FAW has expanded into Southeast Asian countries, including Bangladesh [6] India [7], China [8,9], and Japan. In June 2019, the first invasion of FAW populations in South Korea was confirmed using molecular techniques (cytochrome oxidase subunit I (COI) gene sequencing) [10,11].

FAW has a broad host range of over 180 species [12,13]. Further, it has been globally reported that FAW can be divided into two strains depending on the preferred host plants: the rice strain prefers grass/rice plants, and the corn strain prefers maize, cotton, and sorghum [14]. Changes in insect pest populations are influenced by the nutrition and unique properties of the plants that serve as hosts [15,16]. The varying nutrition of various host plants that insects are reared on throughout their larval stages strongly affects their life history features, such as growth, reproduction, and survival. Population dynamics and the status of pests in the field are significantly influenced by demographic studies [17,18]. Although maize is FAW’s preferred crop [19], other crops can serve as acceptable hosts in the absence of maize crops [20]. Around 70% of corn yield loss in some parts of China and India is caused by the presence of FAW, and research has been conducted on the evaluation of the preference of each variety and the impact on growth [15,21,22].

Insect symbionts not only protect the host from pathogens and parasites, but also increase resistance to pesticides [23]. Insects have different intestinal structures owing to their various anatomical and morphological characteristics, and the intestinal microbiota are diverse among insect species [24]. In particular, the intestinal microflora comprise taxa as abundant as insects; however, it has been shown that bacteria dominate [25]. The composition of the intestinal microflora is influenced by various factors, such as the host environment, diet, and evolutionary, and ecological factors [26]. Insect gut microbiota can not only affect insect fertility, larval development, and lifespan, but also cause behavioral changes [24,27,28,29]. Although insects and gut microbiota interaction have investigated, little is known about their link with host plant preference, showing the need for research on how specific seed/plant-variety-based preferences affect insect microbiome community structure [30,31]. Understanding insect host preference patterns can lead to efficient management techniques due to the increased usage of insecticides and their detrimental effects on both human health and the environment.

The insect microbiota are not only associated with parasite protection but also with a wide range of beneficial host–microbe functions, including food digestion, nutrition, protection, immune response modulation, and detoxification [32]. The behavior of insects is influenced by their gut microbiomes, which may ultimately be used as a novel approach to insect management. However, these strategies heavily rely on a thorough understanding of the microorganisms linked to insects [33]. A number of bacteria may play a significant role in host adaptation and insecticide resistance [34]. For instance, evidence from previous research indicates that Enterococcaceae, notably *Enterococcus* sp., was more prevalent throughout all stages of FAW development [30]. In another study, over 97% Firmicutes, dominated by *Enterococcus* and *Clostridium* sp., were present in late-instar larvae of FAW [35]. The genus *Enterococcus* includes the species *E. mundtii.* The bacterium is a non-motile, cocci or rod shaped, gram-positive, and facultatively anaerobic lactic acid-producing bacteria, but the biological role of *E. mundtii* is currently poorly understood [36]. Due to the complexity and richness of bacteria, it is still largely unknown whether diverse microbes are involved in insect gut function. Recent studies on gut microbiomes have used an agricultural pest, *Spodoptera littoralis,* as an experimental model insect [35]. *S. littoralis*’s gut microbiome has been extensively studied [37]; however, the factors that influence its colonization are unknown.

The aim of this study was to investigate the role of host plant preferences and intestinal microbiota in the larval developmental stages of FAW. Six commonly found varieties of maize, grown in Korea, were selected for evaluation of host preferences, while egg-laying and hatching rates were observed based on the corn varieties. Additionally, the distribution of intestinal microbes for each variety was confirmed through NGS microbiome analysis and culture-based, and gene-specific PCR confirmation. Understanding the interactions between host plant preferences, larval growth, and the prevalence of gut-specific microbiota may help with the future pest control of invasive FAW.

## 2. Materials and Methods

### 2.1. Corn Varieties and Larvae Selection for Assessing the Insect Life Table and Microbiome Analysis

In August 2021, FAW were collected from the experimental field (35°29′26″ N, 128°43′34″ E) of the department of Southern area crop science (Miryang, Republic of Korea), National Institute of Crop Science, Rural Development Administration (RDA), and maintained for the experiment in the insect pest management laboratory of Kangwon National University (Chuncheon, Republic of Korea). Basic rearing conditions and methods were developed according to those previously reported with some modification for FAW [38]. Artificial diet (General Purpose Lepidoptera, F9772, Bio-Serv, Inc., Frenchtown, NJ, USA) was provided during the larval stages.

To compare differences in FAW feeding preferences, development, and gut microbiome among the six corn varieties, we conducted three steps: (1) a laboratory-level life table comparison; (2) a semi-field-level comparison of feeding preferences among varieties in a greenhouse; (3) a field-level comparison and the gut microbiome. In order to minimize experimental errors in each step, the FAW strain with high genetic identity was mass bred by rearing for seven generations in the laboratory, and the eighth generation of eggs and larvae obtained on different laying dates was used for each step of the experiment. Gangwon provincial agricultural research and development institute provided three different categories of the six commercialized varieties of maize (*Zea mays*) corns, including: (i) three waxy corn cultivars (larger size, mebaek 2-ho, whitish color; heukjeom 2-ho, blue color; and meheukchal, purple color); (ii) two yellow-color popcorn cultivars (oryun popcorn, oryun 2-ho); and (iii) one yellow processing corn (Dreamok) (Appendix A).

### 2.2. Investigation of Growth Characteristics/Life Table of Larvae in Laboratory-Level

The hatched first-instar larvae were carefully and individually transferred to a Petri dish (SPL10093 15 × 90 mm^2^; SPL Life Sciences, Pocheon, Republic of Korea). Twenty insect larvae were grouped as a single replication, with a total of three biological replications. Larvae were fed an artificial diet until the third instar. We provided the same varieties of six maize plant’s young leaves and stems after the third instar larvae. Following incubation, all larvae individuals were weighed using a scale, and the weight of the pupae was recorded, as well as the female-to-male ratio for each treatment. The eggs were laid by a single pair mating setting after classifying males and females in an Insect Breeding Dish (SPL310050, 50 × 15 mm^2^, SPL, Pocheon, Republic of Korea). Each variety of young corn leaves and stems was fed daily, and survival, growth, and death were recorded. We fed each maize variety equally at each stage of growth. Statistical analysis was performed using SPSS V22.0 (SPSS Inc., Chicago, IL, USA) to calculate the total survival period, larval period, pupa period, pupa weight, adult period, and emergency ratio, pupation ratio, and egg hatching ratio. Duncan’s multiple range test was used to confirm the significance (*p* ≤ 0.05) of each treatment (Table 1). Larvae were reared in a growth chamber with the specified environmental conditions (25 ± 1 °C, 65.5% relative humidity, and a 16:8 h light: dark photoperiod) [39].

### 2.3. Host Preference Survey and Damage Score Measurement in Green House (a Semi-Field Level Comparison of Feeding Preferences)

Six maize varieties were grown in the greenhouse of the Chuncheon campus, Kangwon National University, in order to understand the host preferences of FAW. Seeds of each variety were sown in a pot with a layer of bed soil (0.5–1 cm) on top (Seoul Bio Co., Ltd., Eumseong, Republic of Korea). The commercialized bed soil composition was zeolite 4, perlite 7, vermiculite 6, cocopeat 68, peat moss 14.73, fertilizer 0.201, wetting agent 0.064%, and pH regulator 0.005%. Furthermore, the composition of physical parameters was moisture content (40–60%), water retention capacity (30–50%), and bulk density (0.15–0.25 Mg/m^3^), and chemical parameters of soil were nitrate–nitrogen (NO_3_-N, 200–350 mg/L), ammonium–nitrogen (NH_4_–N, 150 mg/L), phosphorus pentoxide (P_2_O_5_ (200–350), electrical conductivity (EC) measured as deci-siemens per meter [0.65 (±0.3) dS/m], and pH (1:5 *v*/*v*; 5.5–7.0), as well as other inorganic constituents (K, Ca, Mg, Fe, Cu, Zn, and B). The seeds of each variety were sown in plastic tray (10 × 5 = 50-hole filled, random block design) with bed soil (Appendix A). After two weeks, each corn variety reached a height of approximately 50 cm. The host preferences experiment was conducted in a BugDorm-2400 insect-rearing Tent (75 × 75 × 115 cm^3^) (MegaView Science, Taichung, Taiwan). In each experiment, the 3rd instar larvae of FAW were randomly inoculated in each tent. Larvae were fed with leaves and green stems of the maize corn samples we grew. The damage score was recorded using a standard scoreboard prepared for rearing after seven days (Appendix A). We observed and collected data from each tent (triplicate tent) every 24 h (24, 46, and 72 h for up to seven days) (Table 1). The host preferences for each breed were calculated based on the recorded damage score (0–6; ‘0′ means without damage of plants by inoculated insects, ‘1′ means damage only one place of the entire areas, ‘2′ means damage observed between 10 and 30% of total surface area of the tested plants, ‘3′ means damage observed between 30 and 50%, ‘4′ means over 80% damage, ‘5′ means all leaves except the stem are damaged and ‘6′ means the plant is completely destroyed (100% damage to tested maize plant leaves and stem).

### 2.4. Inoculation of Larvae to Maize Plants to the Agricultural Fields

The maize seeds were randomly planted in a block layout at random in an agricultural field to assess microbial composition of FAW larvae. An equal number of rows and an equal number of seeds were distributed over the natural soil of Kangwon National University agricultural plot in Korea (37°56′20″ N, 127°47′01″ E). The corn varieties were grown from 3 June 2022 to 21 June 2022. When the corn grew to a height of from 50 to 70 cm, the 3rd instar larvae (two individuals) of each plant were inoculated. Immediately after inoculation, the corn was wrapped with a net (70 × 50) to block external factors, and after about two weeks, the corn was cut and the larvae individuals were collected for further assessment of microbial composition. Previous studies have shown that the entire gut can provide a more accurate assessment of the gut’s microbial composition [30,40].

### 2.5. Culture- and Molecular-Based Gut Microbiome Identification

The 5th instar larvae were taken and washed with phosphate-buffered saline (1 × PBS), taken out of the whole intestine, properly macerated with 10 mL of autoclaved distilled water and vortexed. Dissecting tweezers (Dumont #5 1253-21, Fine Science Tools, Heidelberg, Germany) were used to make an incision on the back of the larva’s body to separate the whole intestinal digestive organs.

The gut was dissected from the insects, as previously described [40]. Each sample was homogenized using glass–glass tissue grinder (PYREX™ Glass Pestle Tissue Grinders; Corning, Newark, NJ, USA) in the autoclaved distilled water and serially diluted ranging from 1 × 10^−1^ to 1 × 10^−11^. Lactose agar and broth (LB) and nutrient agar (NA) were used for culturing Enterobacteriaceae, as well as other common bacterial populations. A 100 μL aliquot was spread onto an agar plate. The colony-forming units (CFUs) were counted on each plate after 24–48 h incubation at 30 °C. From each plate, 5–10 distinct morphology bacterial colonies were picked from each plate and deposited at −80 °C with glycerin stock for further study.

For PCR analysis, the pure single bacterial colony was inoculated in Luria Bertan (LB) broth for 35 °C for 18 h in a 5 mL lactose broth (LB) solution. Genomic DNA was isolated from 1 mL of LB culture liquid using DNAzol kit (Molecular Research Center, Cincinnati, Ohio, USA), according to the manufacturer’s instructions.

Afterward, species diagnosis PCR was performed on the species identified as dominant in the result of intestinal microbiome analysis (FAW_Emun_L1 5′-CAAGGCATCCACCGT-3′; FAW_Emun_G1 5′-GAAGTCGTAACAAGG-3′) [41]. The amplified PCR products were electrophoresed on a 1.0% agarose gel and purified using a DNA Gel Extraction Kit (Qiagen, Hilden, Germany). The purified PCR products were sent to Phyzen for sequencing (Phyzen, Seongnam, Republic of Korea).

### 2.6. Nonculture (NGS)-Based Gut Microbiome Analysis

The 5th instar larvae were taken and washed with phosphate-buffered saline (1 × PBS), taken out of the whole intestine. Previous studies have shown that the entire gut can provide a more accurate assessment of gut microbial composition [30,40].

After removing the source of contamination on the body’s surface using sterile water, it was washed once more with larvae phosphate-buffered saline (1 × PBS). An incision was made on the back of the larval body to separate the digestive organs using dissecting tweezers. The entire intestine from five larvae, obtained using a glass–glass tissue grinder using the DNAzol kit (Molecular Research Center, Cincinnati, Ohio, USA), was set as a replication, and a total of three pooled genomic DNA were obtained from FAW larvae reared on each of the six corn varieties.

The sequencing library for metagenomic analysis was constructed according to the illumina 16S metagenomic sequencing library protocol. In brief, the v3–v4 region of 16S rRNA was amplified from 2 ng genomic DNA template with 5× reaction buffer, 1 mM of dNTP mix, 500 nM each of 16S forward (5′-TCGTCGGCAGCGTCAGATGTGTATAAGAGACAGCCTACGGGNGGCWGCAG-3′) and reverse primer (5′-GTCTCGTGGGCTCGGAGATGTGTATAAGAGACAGGACTACHVGGGTATCTAATCC-3′), and Herculase II fusion DNA polymerase (Agilent Technologies, Santa Clara, CA, USA). The cycle condition for the 1st PCR was 3 min at 95 °C for heat activation, and 25 cycles of 30 s at 95 °C, 30 s at 55 °C and 30 s at 72 °C, followed by a 5 min final extension at 72 °C. The PCR product was purified with AMPure beads (Agencourt Bioscience, Beverly, MA, USA) and used for a 2nd PCR with NexteraXT Indexed Primer to construct a library. The library was sequenced using the MiSeq platform (illumina, San Diego, CA, USA) with a 300 bp paired-end (Phyzen).

Feature sequences (amplicon sequence variants, ASVs) were obtained for analysis of the produced nucleotide sequence using Qiime2 [42] (released February 2022), and OTU clustering was performed with 99% identity for ASVs using the search algorithm. Through diversity analysis, alpha rarefaction, PCoA, and UPGMA phylogenetic trees between samples were analyzed, and OTU sequences were compared with the reference genome database to classify which species were distributed in each sample.

### 2.7. Ethics Statement

No national permissions were required to collect samples from public lands. The experiment was conducted at the Kangwon National University agricultural experimental field in Korea (37°56′20″ N, 127°47′01″ E). We did not collect samples from any protected or endangered species.

## 3. Results

### 3.1. Host Preference and Growth Characteristics by Host/Life Table Analysis

Table 1 shows the host preference for FAW of the six corn varieties. Host preference was based on the average of four replications. A significant effect was observed on the larval period, pupal period, egg-hatching ratio, and larvae weight based on the six maize varieties (F (5, 174) = 19.817, *p* = 0.0001; F (5, 174) = 2.41, *p* = 0.039; and F (5, 174) = 13.664, *p =* 0.0001; respectively). The total survival period, adult period, and egg-hatching ratio did not differ significantly among the tested cultivars (F (5, 174) = 1.564, *p* = 0.173; F (5, 174) = 1.156, *p* = 0.333; F (5, 24) = 1.817, *p* = 0. 147 respectively) (Table 1).

### 3.2. Evaluation and Assessment of Damage Score of Host Plant Leaves

We considered a standard damage score of 0–6 for maize plants after seven days of inoculation. Based on the damage performances of FAW insect eating, the ‘B = Heukjeom 2-ho’ corn variety exhibited the highest score and the ‘D = oryun popcorn’ variety exhibited the lowest score (Table 2). However, maize variety ‘B = Heukjeom 2-ho’ had the highest standard damage score (4.50 ± 1.48).

### 3.3. Microbiome Analysis Based on Corn Varieties

To investigate microbial variability, we selected various corn varieties. Eighteen samples were sequenced and analyzed for bacterial communities based on corn varieties. A metagenome sequencing was performed using six maize corn varieties and three biological replicates of insect gut (3 × 6, *n* = 18) samples (MiSeq platform, paired end 2x–250, 16S rRNA v3–v4). After excluding the adapter region, an average of 57,651,180 bp bases were obtained from 18 samples (six varieties, three replications). A total of 216 operational taxonomic units (OTUs) of bacterial 16S rRNA were analyzed from 18 samples, and the average sequence length was 453.4 bp (Appendix A). At the phylum level, Proteobacteria showed the highest number of Firmicutes, followed by Proteobacteria, Actinobacteria, Bacteroidetes, Candidatus Melainabacteria, Deinococcus-Thermus, Fusobacteria, Verrucomicrobia, Gemmatimonadetes, and Cyanobacteria at the top level. Only four families, Firmicutes, Proteobacteria, Actinobacteria, and Bacteroidetes, were prominent compared to the other families (Figure 1A,B).

Firmicutes was the most abundant bacterial community in each tested sample; the dominant bacterial phylum was Proteobacteria, followed by Firmicutes (Figure 1). The dominant phylum in the gut of the larvae of FAW was Firmicutes (94.45% in corn variety ‘A’ 98.80% in ‘F’), followed by Proteobacteria (0.86% in variety ‘A’ 4.91% in ‘A’). As for the bacterial genus that dominates the intestine, *Enterococcus* showed the highest frequency at 83.7–99.1%, followed by *Ureibacillus,* with a maximum of 6.7%. (Figure 1B).

Species diagnostic PCR results also confirmed that *E. mundtii* was dominant in all groups at the species level (90.1–98.4%). In the Shannon index, it was found that the group that consumed Oryun popcorn had the most diverse distribution of intestinal microbes (Figure 2).

Several species of bacteria were investigated in a complex microbial community to find groups of microorganisms that commonly interact (positive/negative relationship) together. Through network analysis, it is possible to determine not only single microorganisms, but also the interdependence of microbial communities that prefer the same habitats or nutrients, as well as how they interact with/affect each other. In network and richness analysis, a positive relationship is shown by a green line, while a negative relationship is indicated by a red line. Line thickness can vary depending on the strength of the relationship (Appendix A).

The 40 most prevalent probable species were chosen and used for the spearman correlation coefficient analysis. *E. mundtii* was the most prevalent bacterial species among the top 40 (Figure 3 and Appendix A). We recovered four *Enterococcus* species through metagenomic analysis and traditional cultural analysis with sequenced with distinct colonies PCR-amplified products from distinct colonies and matched them to the BLAST search, including *E. mundtii*, *E. gallinarum*, *E. casseliflavus*, and *E. rotai* (Table 3). These four species were also exhibited using metagenomic analysis and the spearman correlation coefficient analysis reveals their relationships.

The total viable bacterial count (TVBC) of cultivatable bacteria was expressed as the number of CFU in 1 mL of the sample. The experiments were repeated with different larvae grown under laboratory conditions. The TVBC of the entire intestine was found to range from 6.4 ± 3.08 × 10^8^ to 1.08 ± 0.07 × 10^11^ CFU/mL of FAW larval digestive tract suspension for cultivatable facultative aerobic bacteria. The species level was analyzed with specific primers (intergenic sequence-based primer), sequenced, and matched to the NCBI coverage and similarity analysis (Table 3).

## 4. Discussion

Microorganisms play an important role in the developmental stages of insects [32]. As a major invasive pest, the different developmental stages of corn-feeding in FAW and the microbial community richness and dynamics of the FAW midgut remain unclear. It is widely known that the microbiomes help the insect hosts to digest food, detoxify their ingested insecticides, and to adapt to alternate habitats [32,43]. The nutritional requirements of FAW larvae in their fifth instar are significantly influenced by the bacteria [44]. Gut bacteria contain several enzymes involved in digestion and nutrient intake, indicating that gut microbes may be significant in promoting the efficiency of food utilization in *Spodoptera litura* [45]. Understanding the feeding preference and gut microbial community of FAW is crucial in preventing the spread of invasive pests. A majority of lepidopteran insects consume alkaline plant leaves, including maize, which may vary in pH gradient [46]. The population of the microbial composition is somewhat influenced by the gut environment. Further, the environment of the larvae’s gut is very alkaline to prevent the degradation of toxic secondary compounds in plant leaves. The microbial colonization of the insect gut is a highly dynamic process that is influenced by a wide range of poorly known factors.

We examined host plant preferences during the FAW larval life cycle and the microbial community in the gut of insects based on host preferences and discussed how they interact with the hosts [32]. Firmicutes and Proteobacteria were the predominant bacteria throughout the various developmental stages, although the gut microbiota of FAW varied depending on the different developmental stages. The predominant genus of *Pantoea* and *Citrobacter* of the phylum Proteobacteria were found in early instar larvae [47]. The polysaccharides present in insects can be broken down by this phylum of bacteria [48].

In lepidopteran larvae, the gut microbial community changes as they grow, with *Enterococcus* and *Clostridium* species [18,20] dominating in young larval stages and the family Firmicutes (over 97%) dominating in the mature larval stage before pre-pupal stages. The family Firmicutes was the most (relative abundance 97%) dominant microbiota in the gut of FAW in this study, which was consistent with the findings of Li et al. [30] in FAW, Broderick et al. [49] in *Lymantria dispar*, Priya et al. [50] in *Helicoverpa armigera*, Xia et al. [34] in *Plutella Xylostella* and Chen et al. [51] in *Bombyx Mori.* In a previous study [52], *Enterococcus* spp. (Firmicutes family) were shown to have a similar abundance to the diamondback moth, *P. xylostella* (L.), which supports our dataset (Figure 1). Distinct *Enterococcus* species were identified using cultural assays in 53% of 403 (213/403) insect genera, including the Lepidoptera order, and 100% of the Lepidoptera larval forms produced *Enterococci*. They derived from green and overwintering corn plants. *Enterococci* were identified in insects that were collected in the field during the dormant season, suggesting their potential use as overwintering agents [53].

The common symbiotic bacteria, *Enterococcus gallinarum* and *E. mundtii* comprise the most abundant bacteria in caterpillar hosts and show high tolerance and degradation capacity for organic compounds [54,55]. Further, it has been reported that bacteria can directly degrade organic insecticides, such as ethoprophos, dimethoate, and chlorpyrifos, and these bacteria are often ingested from sources in the environment and human food sources by agricultural pests [56]. Moreover, the gut microbiota may enhance detoxification by influencing host fitness and the immune system [30,37]. A study was conducted by Xia et al. in 2018 [34], showing that *Enterococcus* spp. play an important role in *P. xylostella* insecticide resistance, but the key mechanism for insecticide resistance is not direct. More specifically, *Enterococcus* spp. play a significant role in *P. xylostella* insecticide resistance and the immune system may have a significant impact on how gut microorganisms affect insecticide/pesticide resistance [52].

Field-collected larval populations contain a wider range of microorganisms than laboratory strains. However, the microbial diversity in our metagenomic analysis was observed less often, possibly as a result of the larvae’s brief exposure to external conditions (exposure for 20 days in the field). A previous study showed that the bacteria *E. mundtii* (isolated from *Ephestia kuehniella* larval feces) demonstrated antimicrobial activity against a variety of bacteria, including entomopathogens *Bacillus thuringiensis* and *Pseudomonas entomophila* [57]. Nevertheless, *Enterococcus* produce a wide variety of bacteriocins, called enterocins (small, heat-stable, ribosomally synthesized antimicrobial peptides), that are active against and immune to other bacteria [58]. The variety of maize used to rear the larvae affected the microbiome composition, although information on the variations in composition (nutritional and energy content of used corn varieties) and the dominant chemical components (mainly secondary metabolites) is lacking. We could not establish a relationship between corn varieties and the microbiome composition in reared larvae, which varies depending on the variety of maize used. Although the primary variable was the variety of maize, we were unable to determine what might have contributed to changes in the microbiome’s composition. Furthermore, the environment and nutrition are the key influencing factors for gut microorganisms’ variation [30]. Phenolic compounds are responsible for color (purple, blue, white, and yellow coloration of maize seeds; see Appendix A). Anthocyanins, water-soluble flavonoids, accumulate in the seeds of blue and purple-colored maize. In one study, the purple kernel genotype showed the highest concentration of anthocyanin, whereas yellow maize showed higher content in carotenoids [59]. Like other phenolic compounds, flavonoids have the ability to protect the kernel from biotic and abiotic stresses. The accumulation of phenolic compounds in maize endosperm and pericarp might help the plant resist insect damage. The environment (biotic and abiotic factors), nutrition (amino acid and carbohydrate ratio), and secondary metabolites (phenolic component differences among the six varieties) might be contribute to the differences in gut microbial composition. Microbes also come from the agricultural soil in which we cultivate and spawn our seedlings. In earlier studies, the consumption of tomato leaves (contain soil fungus *Trichoderma afroharzianum*) was shown to possibly alter the symbiotic bacteria of *Enterococcus* and the functional ability of *E. casseliflavus* to provide the insect host with sugars and essential amino acids, which would have a detrimental effect on *S. littoralis’* development and survival [60]. Therefore, future research should be conducted aiming to understand the plasticity of insect gut microorganisms, i.e., insect fitness, soil properties, and composition of the environment in which the host plants are grown, as well as other parameters.

Taken together, insect gut microbes play a crucial role in their metabolic development phase and their immunity. Metagenomics- and culture-based identification methods were utilized in the current work to assess the diversity of microbes and the life table comparability in the gut bacteria of FAW. The diversity of the microbiome varies based on the corn varieties in agricultural field and greenhouse. The most significant differences between individual *Enterococcus* bacteria and other varieties were noted and statistically assessed. However, the gut microbiota are not solely determined by nutrition. According to the culture and metagenome data, a common bacterial species, *E. mundtii,* was detected in the gut of FAW. This species may be responsible for the breakdown of insecticide substances, showing resistance properties. Further research is required to determine whether the species-specific bacteria in such insects are linked to their physiology or to other important biological properties.

## 5. Conclusions

The consumption of six different corn varieties resulted in variations in the gut microbiota and *Spodoptera frugiperda*’s development. Since the plasticity of insect gut microbes helps insects utilize different foods and enhance their fitness, a comprehensive understanding of FAW’s gut microbiome will help in the development of novel pest control strategies for this invasive pest. Further research is needed to explicitly rule out any alternative possibilities (except *Enterococcus*, *Pseudomonas*, or other unknown bacterial species or biotic or abiotic factors that may play a role in insecticide resistance); however, it seems likely that Lactobacillales order or other, rarer-order bacteria play a role in conferring insecticide resistance. Future molecular research on the functions of microbiota communities in the gut or intestine of insects and their potential contribution to the development of host resistance to insecticide should be conducted based on these findings.

## Figures and Tables

**Figure 1 insects-14-00358-f001:**
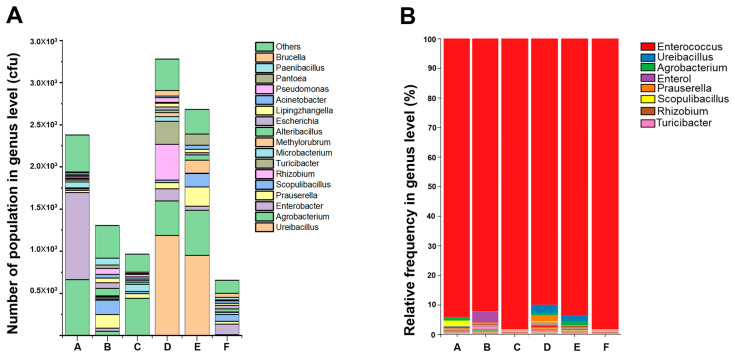
Comparison of the gut microbiota of *Spodoptera frugiperda* larvae fed on each of the six corn varieties. (**A**) shows the number of bacterial populations at the genus level, excluding the highest number of *Enterococcus*, and (**B**) shows the relative abundance of each genus in each of the six corn cultivars. Using metagenome analysis, the different color codes represent various bacterial genera. The figure is based on using six corn varieties [(A) mebaek 2-ho, (B) heukjeom 2-ho, (C) dreamok, (D) oryun popcorn, (E) oryun 2-ho, and (F) meheukchal)]. CFU = Colony forming unit.

**Figure 2 insects-14-00358-f002:**
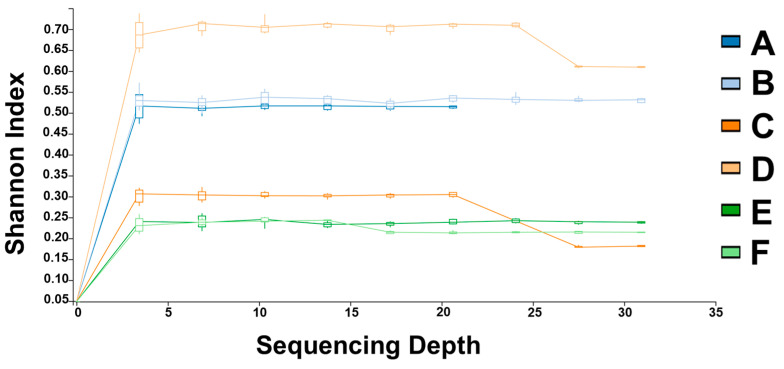
Alpha diversity of cultures at a sequencing depth of 30,000. The Shannon diversity index showed that group D (oryun popcorn) had the most diverse distribution of microbial colonization. Sequencing depth unit is expressed as thousand unit ‘1’ = 1000. Six corn varieties: ((A) mebaek 2-ho, (B) heukjeom 2-ho, (C) dreamok, (D) oryun popcorn, (E) oryun 2-ho, and (F) meheukchal)).

**Figure 3 insects-14-00358-f003:**
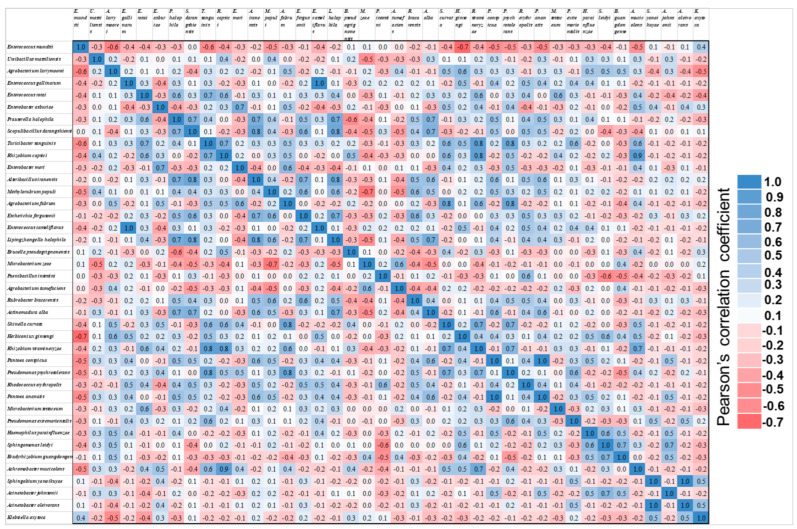
Spearman correlation coefficient quantifying the relationship of linear association between bacterial populations. Results took on a value between −1 and +1, where −1 indicates a perfect negative linear correlation and +1 a perfect positive linear correlation. The value ranged from (−3 to −5) medium, (−5 to −7) strong negative correlation (red color), and the opposite applied to the positive linear correlation (blue color). *Enterococcus mundtii* was the most prevalent bacterial species among the top 40 probable bacterial species. Additional information is exhibited in the Appendix A.

**Table 1 insects-14-00358-t001:** Life table and comparison of growth characteristics based on selected maize (*Zea mays*).

	Cultivar ID (Cultivars)						F Value
	A (Mebaek 2-ho)	B (Heukjeom 2-ho)	C (Dreamok)	D (Oryun Popcorn)	E (Oryun 2-ho)	F (Meheukchal)	
* Total survival period (days)	^†^ 38.77 ± 17.02 ^b^	40.40 ± 10.47 ^ab^	45.50 ± 06.02 ^a^	41.63 ± 10.09 ^ab^	40.83 ± 13.51 ^ab^	44.37 ± 04.89 ^ab^	F (5, 174) = 1.57
							*p* = 0.173
* Larva period (days)	19.57 ± 1.48 ^a^	19.30 ± 1.02 ^a^	19.23 ± 0.57 ^a^	17.50 ± 0.90 ^c^	18.33 ± 0.99 ^b^	18.16 ± 0.74 ^b^	F (5, 174) = 19.82
							*p* = 0.001
* Pupa period (days)	13.26 ±1.14 ^ab^	13.13 ± 1.10 ^ab^	13.64 ± 0.78 ^a^	13.22 ± 1.04 ^ab^	12.77 ± 1.15 ^b^	13.10 ± 1.37 ^ab^	F (5, 174) = 2.41
							*p* = 0.039
* Adult period (days)	13.26 ± 6.22 ^ab^	12.68 ± 5.86 ^ab^	13.50 ± 5.32 ^a^	15.13 ± 6.14 ^ab^	15.32 ± 6.16 ^b^	13.38 ± 4.59 ^ab^	F (5, 174) = 1.16
							*p* = 0.333
Pupa weight (g)	0.22 ± 0.03 ^a^	0.20 ± 0.02 b^c^	0.22 ± 0.02 ^a^	0.20 ± 0.02 ^c^	0.17 ± 0.02 ^d^	0.21 ± 0.03 ^ab^	F (5, 174) = 13.66
							*p* = 0.001
** Emergency ratio (%)	93	83	93	82	88	97	–
** Pupation ratio (%)	100	83	100	97	97	100	–
** Egg hatching ratio (%)	57 ^ab^	32 ^b^	88 ^a^	83 ^a^	73 ^ab^	73 ^ab^	F (5, 24) = 1.81
							*p* = 0.149

^†^ Mean ± SD. Significant differences between several maize varieties are indicated using the identical (uppercase) letters of the alphabet (a, b, c, and d) in each row (significant *p* < 0.01 and *p* < 0.05; one-way ANOVA Duncan test as post-hoc). * Calculated sample size of each cultivar, *n* = 30, entire population of cultivars [*N* (6 cultivars × 30) = 180], ** emergency ratio, pupation ratio, and egg hatching ratio (*n* = 30, entire population and sample size were equal, we calculated percentages only).

**Table 2 insects-14-00358-t002:** The considered damage score based on eating ability for maize plants’ leaves and stems.

Cultivars Name	Cultivar ID	Day 1	Day 2	Day 3	Day 4	Day 5	Day 6	Day 7
Mebaek 2-ho	A *	^†^ 1.06 ± 0.98 ^b^	1.78 ± 1.23 ^b^	2.34 ± 1.36 ^c^	2.78 ± 1.34 ^d^	2.88 ± 1.26 ^d^	3.56 ± 1.48 ^f^	3.88 ± 1.34 ^d^
Heukjeom 2-ho	B	1.22 ± 1.16 ^a^	1.91 ± 1.63 ^a^	2.75 ± 1.68 ^a^	3.28 ± 1.82 ^a^	3.50 ± 1.65 ^b^	4.41 ± 1.56 ^a^	4.50 ± 1.48 ^a^
Dreamok	C	0.94 ± 0.95 ^d^	1.75 ± 1.55 ^c^	2.13 ± 1.52 ^f^	2.53 ± 1.67 ^f^	2.69 ± 1.62 ^f^	3.72 ± 1.80 ^d^	3.81 ± 1.45 ^e^
Oryun popcorn	D	0.97 ± 1.23 ^c^	1.63 ± 1.56 ^d^	2.16 ± 1.72 ^e^	2.59 ± 1.86 ^e^	2.18 ± 1.78 ^e^	3.66 ± 1.93 ^e^	3.81 ± 1.73 ^e^
Oryun 2-ho	E	0.94 ± 0.88 ^d^	1.75 ± 1.16 ^c^	2.28 ± 1.30 ^d^	2.94 ± 1.37 ^c^	3.22 ± 1.29 ^c^	4.00 ± 1.52 ^c^	4.13 ± 1.58 ^c^
Meheukchal	F	0.81 ± 1.00 ^e^	1.63 ± 1.26 ^d^	2.41 ± 1.43 ^b^	3.00 ± 1.55 ^b^	3.34 ± 1.23 ^b^	4.25 ± 1.22 ^b^	4.34 ± 1.49 ^b^

^†^ Mean ± SD. Significant differences between maize varieties are indicated using different letters (uppercase/small letters) of the alphabet (a, b, c, and d) in each column. * Calculated number of each cultivar *n* = 32, Entire population of each cultivar within seven days, N = (7 days × 32) = 192.

**Table 3 insects-14-00358-t003:** Information on main bacterial species from FAW gut microbial culture.

Identify Isolate from FAW Larval Gut	Length (bp)	Coverage (%)	Identity (%)	Ac. No *	Matched to the NCBI (Ac. No.)
*Enterococcus mundtii*	419	99	99	OQ184748	*E. mundtii* strain DSM 4838 (NZ_CP018061.1)
*E. casseliflavus*	415	100	100	OQ184749	*E. casseliflavus* EC20 (NC_020995.1)
*E. casseliflavus*	415	100	100	OQ184750	*E. casseliflavus* EC20 (NC_020995.1)
*E. mundtii*	417	100	100	OQ184751	*E. mundtii* strain DSM 4838 (NZ_CP018061.1)
*E. casseliflavus*	415	100	100	OQ184752	*E. casseliflavus* EC20 (NC_020995.1)
*E. mundtii*	396	100	100	OQ184753	*E. mundtii* strain DSM 4838 (NZ_CP018061.1)
*E. innesii*	415	100	100	OQ184754	*E. innesii* strain DB-1 (NZ_AP025635.1)
*E. casseliflavus*	408	100	100	OQ184755	*E. casseliflavus* EC20 (NC_020995.1)
*E. casseliflavus*	418	99	99	OQ184756	*E. casseliflavus* EC20 (NC_020995.1)

* Ac. No.= The unique accession number provided by NCBI for each submitted sequence

## Data Availability

All available data were produced or analyzed in this manuscript. *S. furgiperda* gut microbiota was deposited in the Sequence Read Archive (SRA) database under accession numbers SRR23078179–SRR23078188 under the Bio Project PRJNA922519. The culture-based bacterial sequences were deposited under the following accession numbers (OQ184748–OQ184756).

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
