# Peer review of "Spodoptera frugiperda (Lepidoptera: Noctuidae) Life Table Comparisons and Gut Microbiome Analysis Reared on Corn Varieties"

_insects, 2023, doi:10.3390/insects14040358_

Round 1

Reviewer 1 Report

The underlying aim of the study is appropriate for this journal and the experiments in general are well-conducted. Therefore, I am recommending minor revision of the manuscript. Specific comments:

Line 46. Both Spodoptera frugiperda and FAW are accepted to name the insects. However the authors use both names interchangeably throughout the manuscript. Please, try to use always the same naming.

Line 49. FAW are present also in Europe and middle East. Please include all the areas where the insect is found.

Line 118. How is perform the selection of the type of larvae? Do the authors have different cohorts in the laboratory?

Line 132. Please include the type of soil where is the seeds potted. In future, it could be useful perform a microbiota analysis of the soil as well, because the microorganisms present in soil may interfere in the plant and therefore in the gut microbiota of the insects (See the manuscript: Di Leilo et al. 2023 https://doi.org/10.1073/pnas.2216922120).

Line 236. Please replace “inoculation” by “inoculation”.

Line 266. Please replace “Figures 2” by “Figures 1 and 2”.

Line 278. Please include the meaning of A to F in the figure caption.

Line 303. The discussion may be improve in several aspects using the manuscript: Di Leilo et al. 2023 https://doi.org/10.1073/pnas.2216922120).

Figure S4. The nodes cannot be read.

Author Response

Thanks for each reviewer's comments and suggestions for improving the present form of this manuscript. We presented each comment point by a point response as below:

Reviewer 1 comments and responses:

Comment 1: “The underlying aim of the study is appropriate for this journal and the experiments in general are well-conducted. Therefore, I am recommending minor revision of the manuscript.”

Response 1:  Thanks for positive comments, we tried to update as your recommendations.

Comment 2: “Line 46. Both Spodoptera frugiperda and FAW are accepted to name the insects. However the authors use both names interchangeably throughout the manuscript. Please, try to use always the same naming”.

Response 2:  Updated

Comment 3:  “Line 49. FAW are present also in Europe and middle East. Please include all the areas where the insect is found”.

Response 3:  Updated (ref no. 4 and 5)

Comment 4:  “Line 118. How is perform the selection of the type of larvae? Do the authors have different cohorts in the laboratory?”

Response 4:   We mention the selection criteria and larvae generation in Line No.112-115.

Comment 5:  Line 132. Please include the type of soil where is the seeds potted. In future, it could be useful perform a microbiota analysis of the soil as well, because the microorganisms present in soil may interfere in the plant and therefore in the gut microbiota of the insects (See the manuscript: Di Leilo et al. 2023 https://doi.org/10.1073/pnas.2216922120).

Response 5:  We mention in details soil types, composition, and properties in Line no. 145-154

Comment 6:  Line 236. Please replace “inoculation” by “inoculation”.

Response 6:  Updated, carefully checked the whole manuscript

Comment 7:  Line 266. Please replace “Figures 2” by “Figures 1 and 2”.

Response 7:  Updated

Comment 8: Line 278. Please include the meaning of A to F in the figure caption.

Response 8:  Updated

Comment 9:  Line 303. The discussion may be improve in several aspects using the manuscript: Di Leilo et al. 2023 https://doi.org/10.1073/pnas.2216922120).

Response 9:  Updated, we discussed based on the recommended references “  Line no. the 422-426.

Comment 10:  Figure S4. The nodes cannot be read.

Response 10:  Updated

Reviewer 2 Report

The publication sent for review does not contain any glaring errors that would disqualify it in terms of its content. However, several problems and questions should be addressed, and the answer to them will undoubtedly increase the value of the work. Also, corrections should be made, mainly editing. Comments and questions:

1.       The subsections in the “Materials and Methods” chapter are illegible. The authors focused on presenting the breeding procedure of maize varieties. However, the information on the relationship between the described plant cultures as a food source for caterpillars is unclear. Was there only one breeding of caterpillars, or were there several of them for different purposes? Were the insects used to describe the life cycle from the same breeding group (on the same plants) as the insects intended for microbiological and genetic analysis? There is no precise information on whether separate breeding of caterpillars was carried out only on one (each separately) specific variety of maize, which was used to analyze the variability of the microbiome, and whether another breeding was dedicated only to the choice test and determining the nutritional preferences of caterpillars (Table 2). If so, were caterpillars taken from this breeding colony for microbiome analysis? A diagram showing the relationships between individual plant and insect breeding and the analyzes performed on them would be beneficial.

2.       In the discussion, the authors focus very much on the variability of the microbiome depending on the pesticides used and also see it as a source of resistance to certain types of pesticides. Despite the indicated citations, however, this analysis is very superficial. Information on specific changes in the microbiome under the influence of specific insecticides and herbicides (including fungicides) would be precious, especially concerning the results obtained. Can antibiotics used in agriculture, especially in animal production, be considered in these considerations?

3.       The authors showed the microbiome's diversity in bred caterpillars depending on the corn variety used for breeding. The dominance of bacteria of the genus Enterococcus was demonstrated. Were the differences between individual microbiomes statistically significant?

4.       Showing differences in the microbiome composition in bred caterpillars depending on the variety of maize used, there is no connection with information about these varieties. The information (the table) with data on the differences in the composition (nutritional and energy values of the corn varieties used) and dominant chemical compounds (mainly secondary metabolites) is a crucial missing element in this publication. The authors do not clearly define what could have been the basis for differences in the composition of the microbiome when the only difference was the variety of corn.

5.       The summary does not refer to the obtained results, which are differences in the life cycle of caterpillars and the microbiome's composition.

Detailed notes:

Often the language of the work is ambiguous or incomprehensible (especially in the M&M chapter). Mental abbreviations are used, readable only by the experimenters. For example: However, the two seeds did not grow in each replicate tray. Finally, insect-reared seedlings… or After seven days, the maize leaves were collected and transferred to a growth chamber (25 ± 1 ËšC, 65 5% relative humidity, and a 16:8 h light: dark photoperiod) for larval feeding, as described below - do the authors mean detached leaves that were fed to insects or whole seedlings. Another example: Each variety of corn leaves and stems was fed daily, and survival, growth, and death were recorded. Were leaves and shoots fed(?).

The numbering of Tables and Figures has been mixed up in the publication - it often does not correspond to the numbers in the header or footer of the text. For example, line 223 - Table S1 (Table 1?); line 238; line 256/258; line 274; row 300.

Particularly often in the M&M chapter, Latin names of bacterial genera and species are not in italics.

Line 56 - wrong citation numbering. It should be [12,13].

Line 62 - Latin names of animals and plants are not inflected.

Line 163 - "spawning" is not the most accurate word. Maybe “lay eggs”.

Line 230 - corn,.maize (?)

Line 232 and 242 - the use of the term "unique" is ambiguous - perhaps more precisely: “the identical (lowercase) letter of the alphabet”; “the different (small) letter of the alphabet”.

Table 2 – what was the number of variables taken for calculations if statistically significant differences were noted in the calculations of the standard deviation almost equal to the (arithmetic?) mean values? In Tables 1 and 2, the values of N and the type of calculated mean should be given.

Figure 1: line 260 - Enetrococcus - should be Enterococcus. In the legend to 2B, there is Enterol (?).

Figure 3 - species names are illegible. Reissue required.

Line 385 - Lactobacillales.

Lines 403-404 - Most lepidopteran insects consume alkaline plant leaves, including maize, which may vary in pH gradient [68] - what does "alkaline plants (maize)" mean? The environment of caterpillar intestinal content is highly alkaline to prevent the decomposition of toxic secondary metabolites in plant food.

Line 423: Further research is needed to explicitly rule out any alternative possibilities; - what alternative possibilities do the authors have in mind?

Figure S4 - text in circles is illegible.

Author Response

Thanks for each reviewer's comments and suggestions for improving the present form of this manuscript. We presented each comment point by a point response as below:

Reviewer 2 comments and questions (responses):

Major comments:

Comment 1:

“The subsections in the “Materials and Methods” chapter are illegible. The authors focused on presenting the breeding procedure of maize varieties. However, the information on the relationship between the described plant cultures as a food source for caterpillars is unclear. Was there only one breeding of caterpillars, or were there several of them for different purposes? Were the insects used to describe the life cycle from the same breeding group (on the same plants) as the insects intended for microbiological and genetic analysis? There is no precise information on whether separate breeding of caterpillars was carried out only on one (each separately) specific variety of maize, which was used to analyze the variability of the microbiome, and whether another breeding was dedicated only to the choice test and determining the nutritional preferences of caterpillars (Table 2). If so, were caterpillars taken from this breeding colony for microbiome analysis? A diagram showing the relationships between individual plant and insect breeding and the analyzes performed on them would be beneficial.”

Response1:

Thank you for your kind and crucial comment. We are fully agreed with your point. So we re-arranged and re-wrote the materials and methods part. Particularly, the following paragraphs have been added to make it easier to understand the overall experimental setup.

To compare differences in FAW feeding preferences, development, and gut micro-biome among the six corn varieties, we conducted three steps: 1) a laboratory-level life table comparison; 2) a semi-field level comparison of feeding preferences among varieties in a greenhouse; 3) a field-level comparison and the gut microbiome. To minimize experimental errors in each step, the FAW strain with high genetic identity was mass bred by rearing for seven generations in the laboratory, and the eighth generation of eggs and larvae obtained by different laying dates was used for each step of the experiment.

Additionally, we newly added a graphical abstract instated of a diagram.

Comments 2:  In the discussion, the authors focus very much on the variability of the microbiome depending on the pesticides used and also see it as a source of resistance to certain types of pesticides. Despite the indicated citations, however, this analysis is very superficial. Information on specific changes in the microbiome under the influence of specific insecticides and herbicides (including fungicides) would be precious, especially concerning the results obtained. Can antibiotics used in agriculture, especially in animal production, be considered in these considerations?

Response 2:

We updated per your recommendations and suggestions. We added limitations and more related discussion based on your comments to our study (line no.396-399) and line no (discussion section line no. 404-428).

Comments 3:  The authors showed the microbiome's diversity in bred caterpillars depending on the corn variety used for breeding. The dominance of bacteria of the genus Enterococcus was demonstrated. Were the differences between individual microbiomes statistically significant?

Response 3:

The reviewers rightly stated the microbial variations based on reared larvae on corn types (six different varieties of maize plants).  In Enterococcus genus, the highest OTU and relative abundance were very high, while the second highest bacteria population was far away. Actually, there's no comparison to others’ genera. The number of Enterococcus genus population in each sample (lowest 19,777 in A and highest 36,889 in E). The genus Enterococcus revealed the highest percentage (83.7 – 99.1%), compared to other genera (0.9 – 16.3%), we can say statistically they are significant.

Comments 4:  Showing differences in the microbiome composition in bred caterpillars depending on the variety of maize used, there is no connection with information about these varieties. The information (the table) with data on the differences in the composition (nutritional and energy values of the corn varieties used) and dominant chemical compounds (mainly secondary metabolites) is a crucial missing element in this publication. The authors do not clearly define what could have been the basis for differences in the composition of the microbiome when the only difference was the variety of corn.

Response 4:

Thanks for the critical observation. We agree wholeheartedly with what you observed. This study solely focused on the host preferences and at the same time we observed the microbial communities with the same group of FAW larvae but in two different separate environments (one was greenhouse bed soil and the other was agricultural field natural soil). So, we could not clearly define what could have been the basis for differences in the composition of the microbiome.  We could not focus on the nutritional composition of seeds and secondary metabolites of plant leaves. These are the limitations, we mentioned in our discussion section (lines no 409-424). In future studies, we will consider it again.

Comments 5:   The summary does not refer to the obtained results, which are differences in the life cycle of caterpillars and the microbiome's composition.

Response 5:

We agree with your observation. We explained in detail in previous comment no. 4.

Minor/ other notes:

Comment 1: Often the language of the work is ambiguous or incomprehensible (especially in the M&M chapter). Mental abbreviations are used, readable only by the experimenters. For example: However, the two seeds did not grow in each replicate tray. Finally, insect-reared seedlings… or After seven days, the maize leaves were collected and transferred to a growth chamber (25 ± 1 ËšC, 65 5% relative humidity, and a 16:8 h light: dark photoperiod) for larval feeding, as described below - do the authors mean detached leaves that were fed to insects or whole seedlings. Another example: Each variety of corn leaves and stems was fed daily, and survival, growth, and death were recorded. Were leaves and shoots fed(?).

Response 1:  Fully organized and re-written of Material and Methods section (Updated)

Comment: The numbering of Tables and Figures has been mixed up in the publication - it often does not correspond to the numbers in the header or footer of the text. For example, line 223 - Table S1 (Table 1?); line 238; line 256/258; line 274; row 300.

Response 1:  Updated

Comment 2: Particularly often in the M&M chapter, Latin names of bacterial genera and species are not in italics.

Response 2:  Updated

Comment 3: Line 56 - wrong citation numbering. It should be [12,13].

Response 3:  Updated

Comment 4: Line 62 - Latin names of animals and plants are not inflected.

Response 4:  Updated

Comment 5: Line 163 - "spawning" is not the most accurate word. Maybe “lay eggs”.

Response 5:  Updated

Comment 6: Line 230 - corn,.maize (?)

Response 6:  Updated

Comment 7: Line 232 and 242 - the use of the term "unique" is ambiguous - perhaps more precisely: “the identical (lowercase) letter of the alphabet”; “the different (small) letter of the alphabet”.

Response 7:  Updated

Comment 8: Table 2 – what was the number of variables taken for calculations if statistically significant differences were noted in the calculations of the standard deviation almost equal to the (arithmetic?) mean values? In Tables 1 and 2, the values of N and the type of calculated mean should be given.

Response 8:  Updated

Comment 9: Figure 1: line 260 - Enetrococcus - should be Enterococcus. In the legend to 2B, there is Enterol (?).

Response 9:  Updated

Comment 10: Figure 3 - species names are illegible. Reissue required.

Response 10:  We agree with you. Further research should be needed for exact species identification. Based on our metagenomic data with software analysis, the putative species (based on the NCBI checking, in which we are unable to confirm the precise species name) were used in interactions amongst each other. We have observed some relationships both positive and negative (-1 to +1). Enterococcus mundtii was the most prevalent bacterial species among the top 40 probable bacterial species. If we change the species name but retain the same Enterococcus genus (four same genera), there will be a debate or a value change.

Comment 11: Line 385 - Lactobacillales.

Response 11:  Updated

Comment 12: Lines 403-404 - Most lepidopteran insects consume alkaline plant leaves, including maize, which may vary in pH gradient [68] - what does "alkaline plants (maize)" mean? The environment of caterpillar intestinal content is highly alkaline to prevent the decomposition of toxic secondary metabolites in plant food.

Response 12:  Updated

Comment 13: Line 423: Further research is needed to explicitly rule out any alternative possibilities; - what alternative possibilities do the authors have in mind?

Response 13:  Mentioned in the conclusion section Line no (L 448-450)

Comment 14: Figure S4 - text in circles is illegible.

Response 14:  Updated